# *Phytophthora* in Horticultural Nursery Green Waste—A Risk to Plant Health

**Kadiatou Schiffer-Forsyth [1], Debra Frederickson Matika [1], Pete E. Hedley [2], Peter J. A. Cock [2] and Sarah Green [1,*]**

[1] Forest Research, Northern Research Station, Roslin EH25 9SY, UK; debbie.frederickson@forestresearch.gov.uk (D.F.M.)

[2] The James Hutton Institute, Dundee DD2 5DA, UK; pete.hedley@hutton.ac.uk (P.E.H.); peter.cock@hutton.ac.uk (P.J.A.C.)

[*] Correspondence: sarah.green@forestresearch.gov.uk

**Abstract:** *Phytophthora* is a genus of destructive plant pathogens. Certain species are damaging to native ecosystems, forestry, and the horticultural sector, and there is evidence of their dissemination in plant imports. Horticultural nurseries are central nodes of the plant trade and previous studies have found a high diversity of *Phytophthora* associated with plant nursery stock. It was subsequently hypothesized that green waste disposal sites in nurseries could harbour diverse *Phytophthora* communities and act as a pathogen reservoir and conduit, facilitating further *Phytophthora* infection of nursery stock and its spread into the wider environment. This project identified *Phytophthora* species associated with green waste at three Scottish nurseries by sampling material from waste piles, water run-off from piles, and roots from discarded plants. Species were identified using a baiting method and sequencing of environmental DNA. Plant nursery green waste was shown to harbour diverse and varied *Phytophthora* species assemblages, with differences among nurseries reflecting biosecurity management practices. Eighteen *Phytophthora* species were detected in the samples, including the highly destructive pathogens *P. ramorum* and *P. austrocedri*. Results suggest that the improved management of waste, for example through effective on-site composting, is essential to reduce the risk of *Phytophthora* pathogens spreading from nurseries into the wider environment.

**Keywords:** *Phytophthora*; plant nursery; accreditation; metabarcoding; baiting; waste management

## 1. Introduction

The genus *Phytophthora* is a group of oomycetes (water moulds) that contains many important plant pathogens infecting and causing economic damage to a broad range of hosts worldwide. They are the causal agents of some of the most destructive tree disease epidemics globally, for example *Phytophthora ramorum* causing 'sudden oak death' in the USA [1], *P. agathidicida* causing kauri dieback in New Zealand [2], and *P. cinnamomi* causing widespread mortality of various woody hosts in Australia, South Africa, and Europe [3]. Of the estimated 500 species in the genus, 180 have been described so far [4,5]. Sixty *Phytophthora* species are known to occur in the UK, with some recent introductions that have become increasingly problematic in the last 20 years, especially as forest pathogens [6]. Due to its wide host range, the genus *Phytophthora* can have wide-reaching impacts across sectors, including ornamental plants in nurseries, natural ecosystems, and horticultural crops. The immense impact of an introduced *Phytophthora* species on food security is best exemplified by potato late blight, caused by *P. infestans*, and the devastating Great Famine that followed its introduction into Ireland and continental Europe in the mid-19th Century [7].

The lifecycle of *Phytophthora* takes place in soil and water, hence the nursery environment is highly conducive to these pathogens. Triggered by waterlogging or rain, the spore-bearing structures (sporangia) release motile zoospores [8] which respond to chemical signals to find a suitable site for infection, usually on fine roots or in the collar region

of plant hosts. Some species can also produce oospores and chlamydospores, persistent structures that can remain dormant for several years [8].

The most effective way to limit the damage of invasive pathogens is to prevent new incursions through interception at the port of entry, but the life cycle of *Phytophthora* means that the pathogen can be dormant and infected plants may not appear diseased. When infected stock arrives at plant nurseries, the proximity to numerous potential host species, high humidity levels, and movement of growing media and water offer favourable conditions for the survival and spread of *Phytophthora* [9,10]. It was with this in mind that the 'Phyto-Threats' project was conducted in the UK, which aimed to characterise *Phytophthora* diversity in plant nurseries in relation to management practices [11,12]. This study, in which 118 nurseries across the UK were surveyed over a three-year period, found 63 *Phytophthora* species in total, including species not previously reported in the UK [12]. It was also observed that plant nursery green waste is often dumped on site and left for extended periods of time, as there is no clear use for this material [11]. In the present study, two detection methods were used to identify species: the sequencing of environmental DNA (metabarcoding) and the baiting of *Phytophthora* into live culture followed by the molecular identification of cultures. These verified methods [13,14] are highly robust and have been used successfully for *Phytophthora* detection in a wide range of environments such as woodlands, public parks and gardens [13–17], and plant nurseries [12,16,18].

The aim of this project was to use these established methods to identify *Phytophthora* species associated with nursery green waste at plant nurseries operating different management practices. *Phytophthora* species assemblages were expected to vary with the overall approach to biosecurity of the nursery, the sample type, and the diagnostic method used. Indeed, diverse *Phytophthora* species assemblages were found, including two recently invasive and highly damaging pathogens, *P. ramorum* and *P. austrocedri*.

## 2. Materials and Methods

### 2.1. Nursery Sampling

Samples were obtained in January and February 2022 from three horticultural nurseries (N1, N2 and N3) located in central Scotland. The sampling method was intentionally biased towards detecting *Phytophthora* and included three different sample types associated with green waste piles: solid waste (spent growing media and partially decomposed plant material); roots from intact dumped plants; and water associated with the waste piles (Figures 1 and 2). A set of management criteria related to plant production and biosecurity was obtained for each nursery (Table 1). Nursery N1 was a non-commercial specialist horticultural nursery that has several biosecurity measures in place, including high-temperature composting of green waste. Nurseries N2 and N3 were commercial nurseries that import a portion of their stock from overseas and dump green waste on site without having a composting system in place (Table 1).

For solid waste sampling, approximately 500 g of a heterogenous mix of spent growing media, stems, leaves and broken-up discarded stock plants was collected from two waste piles per site at three horizons: (A) the upper surface of the pile; (B) mid-way down the pile; and (C) the base of the pile, resulting in six solid waste samples per site. Solid waste samples were split, with 250 g being used for baiting analyses and the remainder being processed for metagenomic sequencing as outlined below. Root samples were also collected in triplicate from plants discarded on the piles which were identifiable to genus or family level. No root samples were taken from N1 because no clearly identifiable plants were present on the waste piles. Thirteen root samples were obtained from N2 and 15 root samples from N3. Approximately 3–5 g of roots was pinched from the base of each root ball and placed into perforated paper envelopes.

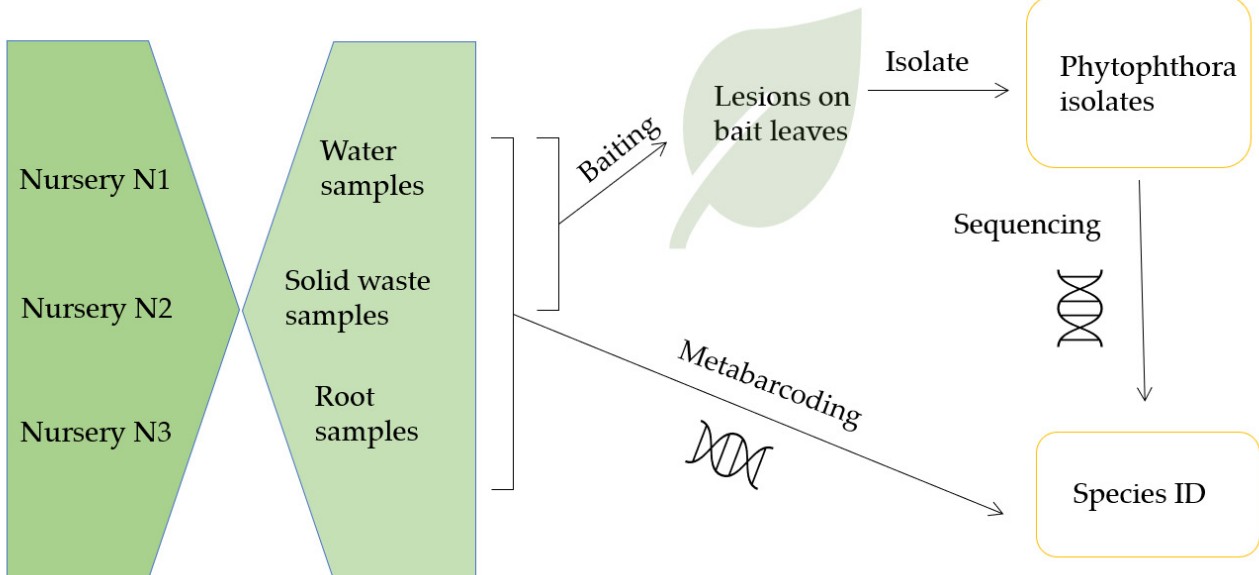

**Figure 1.** Diagrammatic overview of the methods employed for sampling, processing of samples, and analysis.

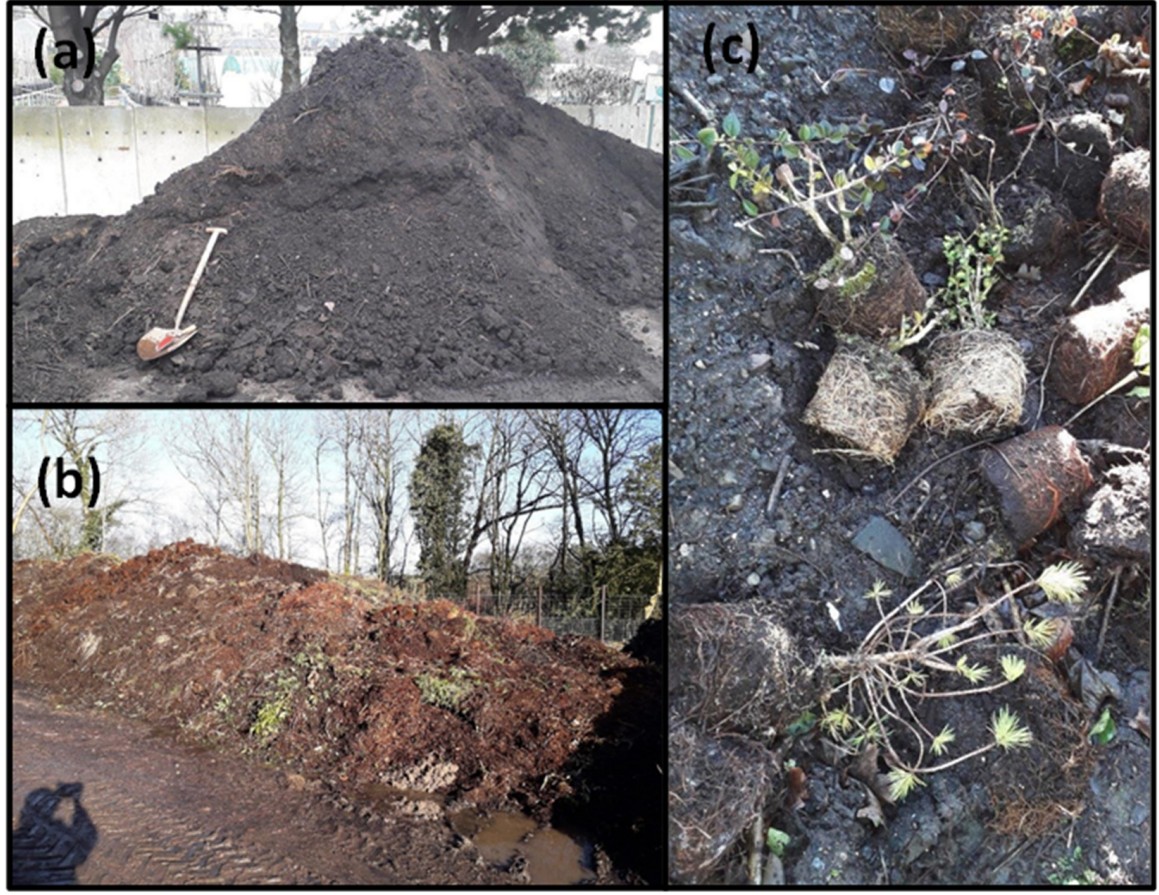

**Figure 2.** Photographs illustrating (**a**) composted waste pile yielding no *Phytophthora*, (**b**) untreated waste pile which yielded multiple *Phytophthora* species including *P. ramorum*, (**c**) discarded plants from which roots were sampled, yielding *Phytophthora*.

**Table 1.** Summary of biosecurity-related management practices in the three surveyed nurseries.

|  | Nursery 1 | Nursery 2 | Nursery 3 |
|---|---|---|---|
| Purpose | Non-commercial specialist horticultural nursery | Commercial nursery | Commercial nursery |
| Irrigation source | Mains | Mains stored in open tank | Stream water in pond |
| % of stock propagated on site | 99 | 0 | 50 |
| % of stock from UK | 1 | 72 | 10 |
| % of stock from EU | 0 | 28 | 40 |
| % of stock from outside EU | 0 | 0 | 1 |
| Waste disposal method | Waste composted on site in concrete holding area | Stored at back of nursery site near stream and hedgerow | Stored at back of nursery site near stream and woodland |
| Other biosecurity measures | Disinfection stations, mats and quarantine area for new stock. | No disinfection stations, mats and no quarantine area. | No disinfection stations, mats and no quarantine area. |

Before sampling water on site, 5 L laboratory mains water was pumped through the equipment described below as a check for contamination (blank lab control). At each nursery, the water used for irrigation was also sampled. Three further water samples were collected per nursery, including puddles draining from waste piles, streams if located less than 10 m away from the waste pile, and a flow-through method in which waste material from the pile was placed into pots on trays and watered to flush out any *Phytophthora* propagules. The flow-through water was left to sit for 30 min in the base of the tray before being sampled. For each nursery water sample collected, 1 L was decanted into a sterile Duran bottle (DWK Life Sciences GmbH, Wertheim, Germany) and processed for baiting. Three 5 L replicate sub-samples of water from each source were then pumped through a 47 mm diameter mixed-cellulose ester filter (Millipore Sigma, Bedford, MA, USA) of 1.2 μm pore size held in a 47 mm polycarbonate in-line filter holder (Pall Corporation, New York, NY, USA) using an adapted knapsack sprayer (CP15 2000 Series Knapsack Sprayer 15 L). Up to three filters for each sample were preserved in 8 mL Longmire lysis buffer [19].

*2.2. Baiting Analyses*

To bait *Phytophthora* from samples, the method described by Pérez-Sierra et al. [20] was followed. Solid waste (250 g) was placed in a sandwich box that was then flooded with sterile distilled water to a level 2–3 cm above the substrate. For water samples, 1 L was placed in each sandwich box. Bait leaves (*Rhododendron* spp., *Hebe* spp., *Hedera helix*, *Quercus suber*, and *Quercus ilex*) were floated on the water surface, the leaves covering as much of the water surface as possible. Controls replicated this set up but with sterile distilled water and bait leaves only. Bait leaves were inspected every day for lesion development. Tissue from lesion margins was surface-sterilised and plated on *Phytophthora*-selective synthetic mucor agar medium [21] and incubated at 16–20 °C. Isolates suspected to be *Phytophthora*, based on the presence of aseptate, hyaline, and freely branching mycelia forming white colonies (Figure 3), were sub-cultured onto V8 agar [22]. *Phytophthora* isolates obtained from baiting were identified with Sanger sequencing of the 900 bp internal transcribed spacer (ITS) ribosomal DNA region amplified using the forward primer ITS6 (5′ GAAGGT-GAAGTCGTAACAAGG 3′) and the reverse primer ITS4 (5′ TCCTCCGCTTATTGATATGC 3′) [23], this being the internationally recognized, standard barcode region for *Phytophthora* identification [24]. To identify the isolate to species level, sequences were analysed with BLAST against GenBank based on a 100% or 99% match to a sequence derived from a type strain or voucher specimen, or published in a peer-reviewed taxonomic paper.

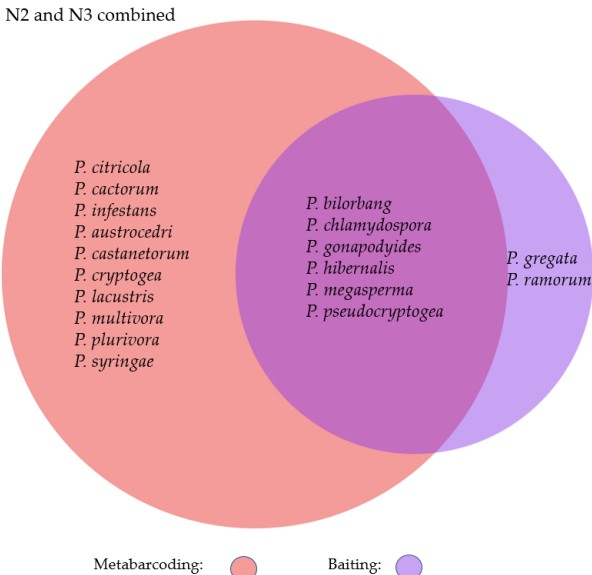

**Figure 3.** Venn diagram showing *Phytophthora* species detected with metabarcoding and baiting at N2 and N3.

### 2.3. Metabarcoding Analyses

For each solid waste sample, 250 g was dried at 60 °C for 72 h and mixed thoroughly. A 20 g aliquot per sample was then milled using a Mixer Mill (MM 400; Retsch GmbH, Haan, Germany) with two 1 cm ball bearings. DNA was extracted from three replicate 250 mg subsamples using the DNeasy PowerSoil Pro Kit (Qiagen, Hilden, Germany) according to the manufacturer's instructions. Root samples were freeze-dried for 24 h, 40 mg per sample milled, and DNA extracted with the DNeasy Plant Pro Kit (Qiagen, Hilden, Germany). For water samples, DNA was extracted in triplicate from 1.5 mL aliquots of the buffer solution in which the filters had been stored, and DNA was extracted using the DNeasy Blood and Tissue Kit (Qiagen, Hilden, Germany) according to the manufacturer's instructions.

Genomic DNA extracted from solid waste, roots, and water samples was processed following the method used by Riddell et al. [15]. The ~260 bp ITS1 region of ribosomal DNA was amplified using nested PCR with the forward primer 18Ph2 (5′ GGATAGACTGTTG-CAATTTTCAGT 3′) and the reverse primer 5.8S-1 (5′ GCARRGACTTTCGTCCCYRC 3′) in the 1st round and the forward primer ITS6 (5′ GAAGGTGAAGTCGTAACAAGG 3′) and the reverse primer 5.8S-1 in the 2nd round, as described in the protocol of Scibetta et al. [13]. Amendments to the protocol include the use of KAPA HiFi HotStart ReadyMix (KAPA Biosystems, Wilmington, MA, USA) and a reaction volume of 12.5 μL, with each reaction containing 4.5 μL PCR-grade water, 6.25 μL Kapa HiFi ReadyMix, 0.375 μL (10 μM) of each forward and reverse primer, and 1 μL DNA or 1 μL round 1 reaction product. Amplification conditions were also altered from the Scibetta et al. protocol [13] with initial denaturation at 95 °C for 3 min (1st and 2nd round), followed by 30 cycles of 98 °C for 20 s, 61 °C for 25 s, and 72 °C for 40 s, with a final cycle of 72 °C for 1 min (1st round) and 25 cycles of 98 °C for 20 s, 61 °C for 25 s, and 72 °C for 25 s with a final cycle of 72 °C for 1 min (2nd round). When PCR amplification was successful, the round 1 product was re-amplified using 2nd round primers with MiSeq adapters that allow Illumina index and sequencing adapter attachment.

Samples were pooled for 250 bp paired-end sequencing on a single flow cell of an Illumina MiSeq sequencer using the MiSeq v. 2 500 bp Standard kit (Illumina, San Diego, CA, USA). The pooled library was loaded at 3 pM, with 40% PhiX control library included. Following quality and error control, and de-multiplexing, each remaining unique sequence was assigned to species using the *Phytophthora* classifier (THAPBI-PICT) developed as part of the Phyto-Threats project and shown to be highly accurate with minimal risk of

false-positive detections [12,25]. Some very closely related *Phytophthora* spp. cannot be distinguished at the ITS1 region, with *P. andina*, *P. infestans*, *P. ipomoeae*, and *P. urerae* being returned as possible identifications for one sequence, for example. These were reviewed manually using known host and geographical range and other factors, in this case leaving *P. infestans* as the most likely source of the DNA. The high sensitivity of amplicon-based PCR means that cross-contamination could result in false positives. To reduce this risk, synthetic controls were included throughout the molecular work, and the bioinformatic pipeline has stringent default abundance thresholds [25].

## 3. Results

Overall, 58 samples were collected from the 3 nurseries (18 solid waste, 12 water, and 28 root samples). Of these, 34.5% were positive for *Phytophthora*, with an average of 2.7 species found per positive sample (Table 2). No *Phytophthora* was detected in the lab blank water controls. No *Phytophthora* spp. were detected at N1, a non-commercial specialist nursery operating high biosecurity standards (Tables 1 and 2). Fifteen *Phytophthora* species were detected at N2 and 9 species at N3; both nurseries are commercial producers of hardy nursery stock (Tables 1 and 2). Eighteen known *Phytophthora* spp. were identified in total across these two nurseries, mostly falling into clades 6 and 8 (Table 2), including two highly destructive clade 8 pathogens: *P. ramorum* and *P. austrocedri*. *Phytophthora ramorum* was baited into live culture from two solid waste samples collected from N2 and from four solid waste samples collected from N3. DNA of *P. austrocedri* was detected with metabarcoding in two water samples (irrigation and stream water) collected from N2 (Table 2). Horizon C, at the base of the waste piles, yielded the highest mean number of *Phytophthora* spp. per sample (2.5) compared with horizons A and B (mean of 0.8 species, respectively).

The two detection methods, metabarcoding and baiting, complemented each other, with six species detected using both methods, ten species detected exclusively with metabarcoding, and two species detected exclusively with baiting (Figure 3). The eight isolates obtained by baiting are shown in Figure 4. Eleven *Phytophthora* spp. were found in the solid waste samples across N2 and N3, including *P. ramorum* and the broad host range pathogens *P. plurivora*, *P. multivora*, and *P. cactorum* (Figure 5). Eight species were found in water samples including two closely related clade 8 species, *P. austrocedri* and *P. syringae*, and five aquatic clade 6 species. Four out of the twelve water samples were positive for *Phytophthora*. Root samples contained five species including the ubiquitous root-infecting pathogens *P. cryptogea* and *P. pseudocryptogea* as well as *P. hibernalis* (Figure 5).

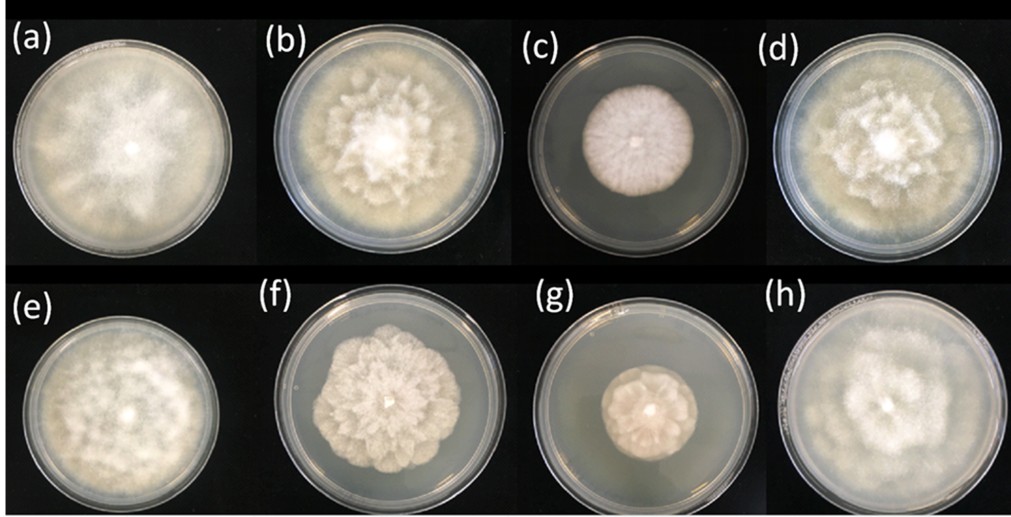

**Figure 4.** Ten-day-old colonies of isolates baited from N2 and N3 growing on V8 agar at 16–20 °C: (**a**) *P. gregata*, (**b**) *P. chlamydospora*, (**c**) *P. ramorum*, (**d**) *P. gonapodyides*, (**e**) *P. pseudocryptogea*, (**f**) *P. hibernalis*, (**g**) *P. bilorbang*, (**h**) *P. megasperma*.

**Table 2.** Summary data of *Phytophthora* species detected in samples associated with green waste in three Scottish nurseries. ND = not detected, NA = not analyzed.

| Site | Sample Type | Species Detected with Baiting, Clade, and Sub-Clade in Brackets | Species Detected with Metabarcoding, Clade, and Sub-Clade in Brackets | Total Number of Species per Sample |
|---|---|---|---|---|
| N1 | 4 water samples 6 solid waste samples No root samples | ND | ND | 0 |
| N2 | Water (on site irrigation water/field blank) | NA | *P. austrocedri* (clade 8) *P. bilorbang* (clade 6b) *P. lacustris* (clade 6b) | 3 |
| | Water (stream bordering nursery site) | *P. bilorbang* (clade 6b) | *P. austrocedri* (clade 8d) *P. lacustris* (clade 6b) | 3 |
| | Root (discarded cypress root ball) | NA | *P. castanetorum* (clade 12) * *P. gonapodyides* (clade 6b) *P. syringae* (clade 8d) | 3 |
| | Root (discarded yew root ball) | NA | *P. cryptogea* (clade 8a) *P. pseudocryptogea* (clade 8a) | 2 |
| | Root (unidentified broadleaf root ball) | NA | *P. pseudocryptogea* (clade 8a) | 1 |
| | Solid waste 1 (horizon A) | *P. chlamydospora* (clade 6b) | ND | 1 |
| | Solid waste 1 (horizon B) | ND | *P. citricola* (clade 2c) *P. plurivora* (clade 2c) *P. chlamydospora* (clade 6b) *P. castanetorum* (clade 12) * | 4 |
| | Solid waste 1 (horizon C) | *P. ramorum* (clade 8c) *P. pseudocryptogea* (clade 8a) *P. chlamydospora* (clade 8a) *P. hibernalis* (clade 8c) | *P. syringae* (clade 8d) *P. plurivora* (clade 2c) | 6 |
| | Solid waste 2 (horizon A) | ND | *P. megasperma* (clade 6b) | 1 |
| | Solid waste 2 (horizon B) | ND | *P. syringae* (clade 8d) *P. castanetorum* (clade 12) * *P. cactorum* (clade 1a) | 3 |
| | Solid waste 2 (horizon C) | *P. chlamydospora* (clade 6b) *P. gonapodyides* (clade 6b) *P. ramorum* (clade 8c) *P. gregata* (clade 6b) | *P. syringae* (clade 8d) | 5 |
| N3 | Water (puddle) | NA | *P. infestans* (clade 1c) *P. chlamydospora* (clade 6b) *P. gonapodyides* (clade 6b) *P. syringae* (clade 8d) | 4 |
| | Water (puddle) | *P. chlamydospora* (clade 6b) | ND | 1 |
| | Root (discarded pine root ball) | NA | *P. hibernalis* (clade 8c) | 1 |
| | Solid waste 3 (horizon A) | *P. gonapodyides* (clade 6b) | ND | 1 |
| | Solid waste 3 (horizon B) | *P. ramorum* (clade 8c) | ND | 1 |
| | Solid waste 3 (horizon C) | *P. chlamydospora* (clade 6b) *P. gonapodyides* (clade 6b) *P. ramorum* (clade 8c) | *P. cactorum* (clade 1a) *P. syringae* (clade 8d) | 5 |
| | Solid waste 4 (horizon A) | *P. chlamydospora* (clade 6b) *P. gonapodyides* (clade 6b) *P. ramorum* (clade 8c) | ND | 3 |
| | Solid waste 4 (horizon B) | *P. chlamydospora* (clade 6b) *P. megasperma* (clade 6b) *P. gonapogyides* (clade 6b) | *P. multivora* (clade 2c) *P. syringae* (clade 8d) *P. gonapodyides* (clade 6b) | 5 |
| | Solid waste 4 (horizon C) | *P. chlamydospora* (clade 6b) *P. ramorum* (clade 8c) | ND | 2 |

* Jung et al. [26] placed this recently described species into a proposed clade 12.

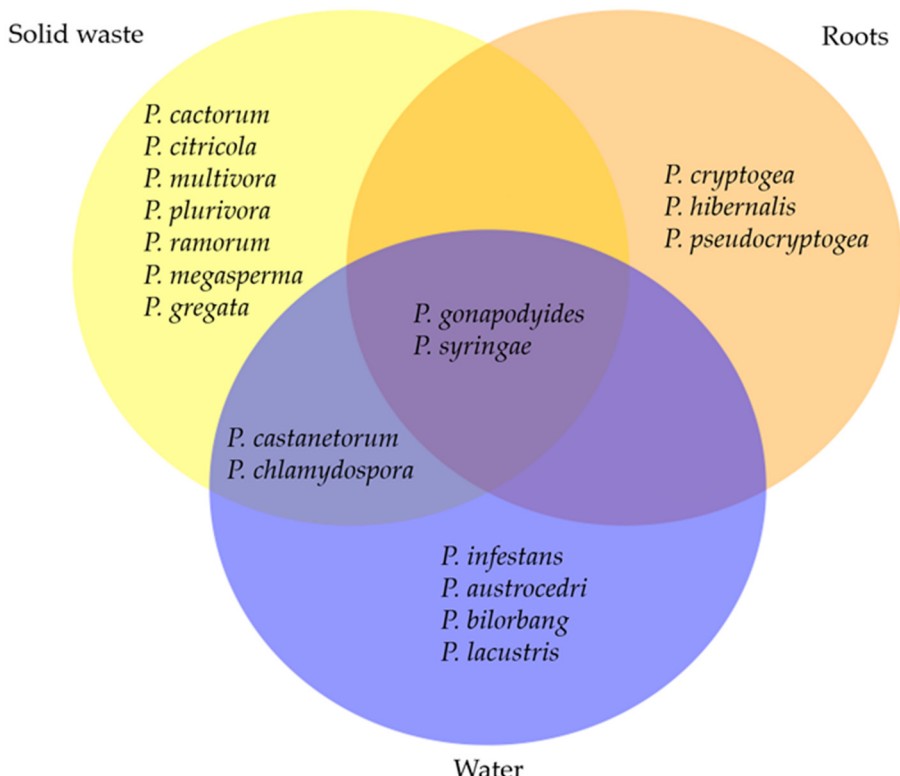

**Figure 5.** Venn diagram showing *Phytophthora* species detected in different sample types at N2 and N3.

## 4. Discussion

A surprisingly high diversity of *Phytophthora* spp. was detected in waste piles at two of the three nurseries sampled in this study, the majority of them being well-studied pathogens on a range of woody and non-woody plant species grown for the ornamental trade. One key finding was the live culturing of several isolates of the quarantine-regulated pathogen *P. ramorum* from [27] nurseries N2 and N3. This pathogen, which has a known host range exceeding 100 different plant species, including many commonly traded ornamental taxa such as *Rhododendron*, *Camellia*, and *Viburnum* [12], is currently the cause of devastating epidemics of oaks in the western USA and of larch in the UK [28,29]. Wider environment epidemics such as these are thought to have been initiated by inoculum originally introduced on imported planting material [15]. *Phytophthora ramorum* was predominantly found in the lowest horizon of solid waste piles, which also harboured the highest number of other *Phytophthora* species. Since *Phytophthora* requires water for dispersal of motile zoospores, the high water content at the bottom of the waste piles allows it to leach through the waste material, raising the risk of spread of these pathogens in water run-off from the piles. Also of concern is the detection of *P. austrocedri* DNA in stream and irrigation water at nursery N2. This pathogen is quarantine-regulated and invasive in the wider environment, causing widespread mortality of native juniper (*Juniperus communis*), an important ecological species in Britain [30]. *Phytophthora austrocedri* is also frequently detected on roots and foliage of nursery stock of other Cupressaceae hosts including *Cupressus* x *leylandii* and *Chamaecyparis lawsoniana* [12], again raising concerns about its global spread in the nursery trade.

Other aggressive *Phytophthora* pathogens found to be present in or associated with the nursery waste piles included *P. cryptogea* and *P. pseudocryptogea* [31], both of which are cosmopolitan species globally, causing damping off and root rot of plants in well over 100 genera [8], and *P. cactorum* [32], *P. citricola* [33], *P. multivora* [34], *P. plurivora* [33] and *P. syringae* [35]. All these species are considered common in Britain, causing root, stem, and shoot cankers on many plant hosts [36,37]. DNA of *P. castanetorum* was detected in

roots and solid waste samples at N2. This species was first described from *Castanea sativa* in Portugal and is considered to be a fairly weak pathogen which may be endemic to Europe [26]. *Phytophthora hibernalis* was baited into culture from solid waste at N2 and detected with metabarcoding in a pine root at N3. This species was originally described on citrus in Australia [38] and is regarded as having a limited distribution in the wider UK environment [39], thus its finding here is of concern as it suggests dissemination through the nursery trade. Other species detected include the aquatic clade 6 *Phytophthora*s, such as *P. gonapodyides*, *P. chlamydospora*, and *P. megasperma*, which are regarded as native and less pathogenic [4] although all three of these species have been found causing lesions on riparian trees in Britain [39]. The baiting into live culture of the other clade 6 species *P. bilorbang* and *P. gregata* is of interest as detections of these species in the UK to date have been rare. Notably, *P. bilorbang* has recently been found causing damage to olive trees in Italy [40]. *Phytophthora infestans*, the causative agent of potato late blight, was detected in water from a puddle at N3, along with three other species. This mix of species present in plant nurseries is concerning, because it may facilitate hybridizations and the development of more virulent pathogenic species and strains [41].

Differences in the results obtained from the two methods used to detect *Phytophthora* spp. could be due to several methodological reasons. Baiting exploits the selective pathogenicity of *Phytophthora* for living tissue and is an effective method for isolating *Phytophthora* from environmental samples [4,20]. Eight isolates were obtained from baiting and could unambiguously be identified using Sanger sequencing. Baiting also has the advantage of providing objective evidence that a species is present and viable, thus able to infect host plants. *Phytophthora* spp. that are weaker competitors, slower-growing, or that do not readily produce zoospores under laboratory conditions can be outgrown by a few stronger competitors and the number of species detected with baiting may be artificially low [16]. However, the baiting of slower-growing species *P. ramorum* and *P. hibernalis*, as well as fast-growing clade 6 species, suggests that the baiting protocol used here worked well. Comparatively, metabarcoding offers a broader and more complete overview of species present as it is extremely sensitive, detecting resting spores and DNA that may not necessarily originate from living propagules. *P. ramorum* and *P. gregata*, however, were detected with baiting but not with metabarcoding in this study. This could be due to the fact that *Phytophthora* propagules are unevenly distributed in the substrate, and that for metabarcoding DNA was extracted from only 750 mg (250 mg in triplicate) of sample substrate, compared with 250 g sample substrate used for baiting. The discrepancy in species detected between methods due to the heterogenous distribution of propagules in soil or bulk substrate is also discussed by La Spada et al. [16], highlighting the complementarity of the two methods and advantages of their use in combination.

The findings of this project tend to reflect the biosecurity management practices of each nursery and therefore provide an evidence base for improving nursery waste disposal methods. Both N2 and N3, which had abundant *Phytophthora*s associated with their green waste, are commercial growers which import live plants and discard their green waste onto piles adjacent to each nursery. These waste piles are not properly composted and are located next to streams and hedgerows which may potentially facilitate pathogen spread into the wider environment. N1, conversely, had no *Phytophthora* detected in the waste piles with baiting or metabarcoding. This nursery does not import live plants and operates stringent biosecurity practices, including a high-temperature composting system for green waste which is located in a concrete holding area away from plant stock and hedgerows. The results of this project, combined with the insights from previous research [9–12], suggest there is likely to be a strong causal link between more stringent biosecurity measures and lack of *Phytophthora* detected at N1. Results from this work were communicated to nursery managers along with recommendations on how to improve practice, and in response N2 has planned to replace their waste pile with a composting system and to spread the fully composted material on agricultural land.

At the time that this study was conducted, there was no sector-wide approach to nursery green waste management in the UK and no clear advice for plant nursery managers on how to deal with waste from a plant health perspective. Elliot et al. [42] surveyed a range of nursery stakeholders in the UK to understand their waste management practices and perceptions on associated plant health risks. The majority of respondents dealt with their waste by dumping on site, yet most (73%) agreed with the need to implement changes on their nursery in regard to plant waste management and biosecurity, with the main limiting factors being lack of guidance on available options and concerns over high costs.

Aiming to address this issue, and utilizing evidence from this diagnostic study, guidance on biosecurity best practice for safe disposal of plant waste and spent growing media has now been published by Elliot et al. in the form of a flier which is available online [42]. The guidance advises growers to minimize waste and risk of infected waste material by growing clean plants, provides advice on safe waste storage prior to disposal, and presents information on options for safe disposal of green waste including on-site composting, incineration, disposal to landfill, or removal to a commercial composting facility. In the UK, the Plant Health Management Standard (PHMS) underpins the 'Plant Healthy Certification Scheme' currently being rolled out for horticultural businesses [43] with the aim of widespread adoption across the ornamental and amenity horticulture supply chain. Businesses and organizations that join the scheme will have to comply with the PHMS and will be identifiable as those which handle plant material in a manner that promotes plant health and biosecurity, including low-risk waste management practices. Given the ubiquity of *Phytophthora* in horticultural trade networks across Europe and globally [9,10], it is hoped that evidence presented here on biosecurity risks from plant waste will help to inform management practices beyond the UK.

**Author Contributions:** Conceptualization, S.G. and D.F.M.; methodology, S.G., D.F.M., P.J.A.C. and P.E.H.; software, P.J.A.C.; validation, P.J.A.C.; formal analysis, P.J.A.C. and P.E.H.; investigation, K.S.-F.; data curation, K.S.-F.; writing—original draft preparation, K.S.-F.; writing—review and editing, S.G., D.F.M. and K.S.-F.; visualization, K.S.-F.; supervision, S.G. and D.F.M.; funding acquisition, S.G. All authors have read and agreed to the published version of the manuscript.

**Funding:** This work was commissioned by Scotland's Centre of Expertise for Plant Health Funded by Scottish Government through the Rural & Environment Science and Analytical Services (RESAS) Division under grant agreement No PHC2021/02.

**Data Availability Statement:** The raw Illumina reads of data presented in this study are openly available from Zenodo at (https://doi.org/10.5281/zenodo.7925609) as FASTQ files (accessed on 10 April 2023).

**Acknowledgments:** The authors wish to thank the three partner nursery managers for allowing us to sample on their premises, enabling this project to go ahead. We also thank Jenny Morris for technical support with the metabarcoding at the James Hutton Institute.

**Conflicts of Interest:** The authors declare no conflict of interest.

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
