# Peer review of "Phytophthora in Horticultural Nursery Green Waste—A Risk to Plant Health"

_horticulturae, doi:10.3390/horticulturae9060616_

Round 1
Reviewer 1 Report
In this manuscript (horticulturae-2399076) entitled " Phytophthora in horticultural nursery green waste - a risk to plant health" submitted to Horticulturae, Kadiatou Schiffer-Forsyth and colleagues have identified Phytophthora species associated with green waste at three Scottish nurseries by sampling material from waste piles, water run-off from piles and roots from discarded plants. Plant nursery green waste was shown to harbour diverse and varied Phytophthora species assemblages, with differences among nurseries reflecting biosecurity management practices. Eighteen Phytophthora species were detected in the samples, including the highly destructive pathogens P. ramorum and P. austrocedri. Authors suggest that the improved management of waste, for example through effective on-site composting, is essential to reduce the risk of Phytophthora pathogens spreading from nurseries into the wider environment. Overall, I consider this research is interesting, but this present version is unsuitable for publication.
1, In this study, three Scottish nurseries were sampled. Theses nurseries should be representative, and their detailed location should be described in the revised version.
2, Authors focused on the Scottish nurseries in this study, and this key information should be clearly included into the revised title.
3, Effects of improved management of waste (e.g. effective on-site composting) on the risk of Phytophthora pathogens spreading should be analyzed in the revision.
4, In this study, authors analyzed the solid waste samples, and the water waste samples should be included in the revised manuscript.
5. Please double-check the reference list. For instance, the full name but not abbreviations of journals are incorrectly presented.
Minor editing of English language required
Author Response
Reply to Reviewer 1 Comments
1, In this study, three Scottish nurseries were sampled. Theses nurseries should be representative, and their detailed location should be described in the revised version.
Response 1: The three nurseries chosen for sampling are representative of management practices found throughout the sector. Detailed locations of nurseries can unfortunately not be disclosed, as there is a confidentiality agreement with the nursery managers that allowed us to take samples in their nurseries.
2, Authors focused on the Scottish nurseries in this study, and this key information should be clearly included into the revised title.
Reply 2: The sampled nurseries are located in Scotland; however, the nurseries sampled are representative of horticultural enterprises in the UK and in other countries. Further, the issue of Phytophthora in plant nurseries highlighted in this paper is more generalized and not limited to Scotland, see for example Jung et al. 2016 (https://doi.org/10.1111/efp.12239).
3, Effects of improved management of waste (e.g. effective on-site composting) on the risk of Phytophthora pathogens spreading should be analyzed in the revision.
Reply 3: This is a valid point but testing material before and after composting in order to assess the effectiveness of the process for killing pathogens was beyond the scope of this study. Nonetheless it was noted that N1 had an effective on-site composting set-up, where material is kept on a hard standing, regularly turned and the temperature monitored and no Phytophthora were found. From this study alone, it is not possible to link composting and absence of Phytophthora here, because other biosecurity measures, such as growing the vast majority of plants from seed on site could have a stronger effect for N1.
4, In this study, authors analyzed the solid waste samples, and the water waste samples should be included in the revised manuscript.
Reply 4: There is no water waste produced by the activities of the nurseries, but runoff associated with waste piles was included in the study, either present as puddles and streams, or artificially created for sampling with the flow-through method as described in the paper. Water samples are included in Table 2 are discussed in the manuscript. Edit made for clarification line 203 “Four out of the twelve water samples were positive for Phytophthora.”
- Please double-check the reference list. For instance, the full name but not abbreviations of journals are incorrectly presented.
Reply 5: Thank you, the reference list has been checked and corrected.
Reviewer 2 Report
The current communication article entitled “Phytophthora in horticultural nursery green waste – a risk to plant health” by Schiffer-Forsyth et al. highlights the role of oomycetes (Phytophthora) contamination in horticultural nursery green waste and its potential impact on plant health. The paper addresses the importance of identifying the presence of Phytophthora in green waste and its potential impact on the health of the plant, which is critical to the success of the horticulture industry. After a careful reading, I found this manuscript interesting and timely. The paper is well-organized, and the author provides a comprehensive review of existing literature on Phytophthora in horticultural nursery green waste. The author highlights the different types of Phytophthora species and their impact on plant health. The paper also discusses the sources of Phytophthora contamination and the methods used to detect and control its spread. However, I have major concerns regarding the English of the paper which needs to be scrutinized by a native writer. The paper can be accepted for publication in the Horticulturae journal after the following moderate revisions:
1. Please consider revising the title to: Phytophthora (oomycetes ) in Horticultural Nursery Green Waste: A Threat to Plant Health.
2. The authors discussed the UK context which is fine, but what about the global status of Phytophthora? Please add this information in the introduction.
3. The basic information about Phytophthora in terms of its early historic detection and nomenclature is missing. Please discuss why it was previously considered fungi and then categorized into a separate entity “oomycetes”.
4. Enrich the introduction with a more focused hypothesis (biochemical interaction between plant and oomycetes) based on recent studies.
5. Tables: Please change to a three-line format of MDPI.
6. Line 55: please replace “project” with “study”.
7. Line 59: what kind of environments? Shift it to the previous paragraph. The last paragraph should indicate the clear objectives and novelty of the study only.
8. While writing “species”. It should be “species” at first mention but “spp.” after this.
9. Please provide some study photographs showing the sampling of horticultural waste and Phytophthora infection.
10. Undefined abbreviations (DNA, ITS, SMA, etc.) should be defined at their first mention.
I have major concerns regarding the English of the paper which needs to be scrutinized by a native writer.
Author Response
Reply to Reviewer 2 Comments
- Please consider revising the title to: Phytophthora (oomycetes ) in Horticultural Nursery Green Waste: A Threat to Plant Health.
Reply 1: Thank you, we think that including this in the title is not necessary as this is mentioned in the first sentence of the introduction.
- The authors discussed the UK context which is fine, but what about the global status of Phytophthora? Please add this information in the introduction.
Reply 2: More information about the global status of Phytophthora has been added to the introduction. Edits made lines 26 to 32.
- The basic information about Phytophthora in terms of its early historic detection and nomenclature is missing. Please discuss why it was previously considered fungi and then categorized into a separate entity “oomycetes”.
Reply 3: We think that this information is not relevant here, as this is not a taxonomic paper that discusses the evolution of the genus. The introduction has intentionally been kept brief to get to the point of this study.
- Enrich the introduction with a more focused hypothesis (biochemical interaction between plant and oomycetes) based on recent studies.
Reply 4: We are not quite sure what is being asked for here. It is well known that Phytophthora infect plants and there are many papers about the biochemical interaction between plants and oomycetes. This was not the focus of the diagnostic study described in this manuscript.
- Tables: Please change to a three-line format of MDPI.
Reply 5: Edits made to Tables 1 and 2.
- Line 55: please replace “project” with “study”.
Reply 6: Edit made on line 55
- Line 59: what kind of environments? Shift it to the previous paragraph. The last paragraph should indicate the clear objectives and novelty of the study only.
Reply 7: Edit made to clarify what environments are meant line 62 and this has been moved to the previous paragraph. The last paragraph of the introduction has been edited to include clear objectives and main the findings from the study.
- While writing “species”. It should be “species” at first mention but “spp.” after this.
Reply 8: Yes, thank you. Edits have been made in relevant places.
- Please provide some study photographs showing the sampling of horticultural waste and Phytophthora infection.
Reply 9: An additional figure (Figure 2) is included in the revised manuscript showing the waste piles and discarded plants.
- Undefined abbreviations (DNA, ITS, SMA, etc.) should be defined at their first mention.
Reply 10: We think that DNA does not need to be defined, the internal transcribed spacer (ITS) is defined at first mention on line 120 and SMA has been replaced with synthetic mucor agar line 117.
Reviewer 3 Report
The authors have addressed the issue of the increase in the number of Phytophthora species in the horticulture green waste in the nurseries across the UK. The study was performed on three nurseries from central Scotland with one being noncommercial (N1) and the other two for commercial purposes (N2 and N3). The samples were taken from solid waste, root, and water waste. They have used two different methods for identification of the species: metabarcoding and baiting. Their results confirmed that both detection methods were efficient in the detection of 6 species of Phytophthora. They found diverse numbers of Phytophthora species in the waste of N2 and N3. Overall experiments are well performed and explained in the text. The results support the conclusions made in the manuscript. However, authors have not addressed whether other factors might also contribute to the presence of Phytophthora species. The result section is very short with no further analysis of the identified species.
Comments:
1. Add a brief major conclusion of the study at the end of the Introduction.
2. Does the time of the year sampling was done play a role in the distribution of species?
3. It would be good if the authors provided a more detailed analysis of the species, the result section i.e., a figure showing their relatedness etc.
The quality of English language was good and did not find any major issues
Author Response
Reply to Reviewer 3 Comments
- Add a brief major conclusion of the study at the end of the Introduction.
Response 1: A sentence has been added at the end of the introduction stating the main conclusions from the study. Edit made line 65 to 68.
- Does the time of the year sampling was done play a role in the distribution of species?
Response 2: It is possible that the time of year may play a role in the distribution of species in environmental sampling. However, in this human-made and highly disturbed habitat that is plant nursery green waste, we suspect that other factors are more important than seasonality. Moreover, metabarcoding can detect DNA from dead or dormant Phytophthora species present in the material.
- It would be good if the authors provided a more detailed analysis of the species, the result section i.e., a figure showing their relatedness etc.
Response 3: To provide more information on the species found and their relatedness, we have included the clade and subclade of each species in table 2. As this is not a taxonomic study, a phylogenetic tree would not have added to the results and this information can be found in other papers cited (Yang et al. 2017).